# Oxyalkylation of Lignoboost™ Kraft Lignin with Propylene Carbonate: Design of Experiments towards Synthesis Optimization

**DOI:** 10.3390/ma15051925

**Published:** 2022-03-04

**Authors:** Fernanda Rosa Vieira, Ana Barros-Timmons, Dmitry Victorovitch Evtuguin, Paula C. O. R. Pinto

**Affiliations:** 1Department of Chemistry, CICECO-Institute of Materials, Campus de Santiago, University of Aveiro, 3810-193 Aveiro, Portugal; dmitrye@ua.pt; 2RAIZ, Forest and Paper Research Institute, Quinta de S. Francisco, 3801-501 Aveiro, Portugal; paula.pinto@thenavigatorcompany.com

**Keywords:** lignin-based polyol, oxyalkylation process, optimization, viscosity, hydroxyl number, polyurethane

## Abstract

Oxyalkylation with propylene carbonate (PC) is a safe process to convert lignin into a reactive liquid polyol to be used in polyurethane formulations. In this study, the effect of operating conditions of oxyalkylation (temperature, time and quantify of PC) on the quality of lignin-based polyol in terms hydroxyl number (I_OH_) and viscosity was studied. Full factorial modeling and response surface methodology (RSM) were applied to study the effect and interaction of process variables on the I_OH_ and viscosity of lignin-based polyols. The results revealed that the I_OH_ is highly affected by the reaction time, while the viscosity is affected by the amount of PC. Validation experiments confirmed the model is reliable. Furthermore, RSM optimization allowed to reduce the amount of PC by about 50% and to increase the lignin content in the polyol from 12.5% to 25% (*w*/*w*) depending on the temperature and time of the process and also on the purpose of the polyol produced (i.e., application in rigid foams or adhesives).

## 1. Introduction

In recent years, the production of bio-based polyols has attracted great attention since conventional polyols are produced from petrochemical derivatives, which make a significant contribution to the increase of greenhouse gas emissions. Bio-based polyols are currently available on the market and are supplied by companies such as BASF SE, Bayer, Dow Chemical, Huntsman, Covestro, Cargill, Dupont and Roquette. However, these companies are producing bio-based polyols mainly from vegetable oils and sugar platforms competing with food applications [1,2,3]. Another potential candidate for the production of bio-based polyols is kraft lignin, a by-product of the pulp and paper industry, and widely available from black liquor (about 55–90 million tons of kraft lignin/year), which is mainly burned for energy supply [4]. Lignin is a recognized natural source of polyols due to the presence of phenolic and aliphatic hydroxyl groups (OH) in their structure. However, achieving the functionalization of these hydroxyl groups, especially the phenolic OH group, with high selectivity is a remarkable challenge due to steric constraints, recalcitrance and the highly variable chemical structure of lignin. Therefore, chemical modifications to convert it into liquid polyols to produce polyurethanes (PUs) are necessary [5].

In general, the most promising route that has been used to convert lignin into liquid polyol is oxyalkylation. Many studies explored the oxyalkylation of different types of lignin with alkylene oxides (e.g., propylene oxide), which became sufficiently liquid to be crosslinked with isocyanates to yield PUs [6,7,8,9,10]. Yet, propylene oxide (PO) is the main building block to produce petrochemical polyols such as polyether. Furthermore, the low levels of process safety associated with the low boiling point of PO (34 °C) and high vapor pressure, which may lead to uncontrolled highly exothermic chain reactions, is a major constraint to scaling up this process. In turn, cyclic carbonate compounds are an alternative to PO since they are compounds with low toxicity and vapor pressure and high boiling point (242 °C) and can easily react with amines, alcohols and carboxylic acids, to initiate ring-opening polymerization under atmospheric conditions [11,12]. Kühnel et al. [13] successfully carried out the oxyalkylation of organosolv lignin with propylene carbonate (PC) to afford polyether polyols. The group’s subsequent study showed that the parameters of the process such as catalyst, catalyst/lignin ratio, reaction time, temperature and PC amount have a great impact on the conversion of solid lignin into liquid polyol. Under optimized reaction conditions using 1,8-diazabicyclo[5.4.0]undec-7-ene (DBU) as catalyst, a degree of substitution (DS) of 0.92 and chain length of up to 4.6 propyl units of oxyalkylated organosolv lignin were achieved by the same group [14]. After that, Maiorana et al. patented the process to produce polyurethane using oxyalkylated lignin from cyclic carbonate and Xuefeng et al. have reported the oxyalkylation of kraft lignin using ethylene carbonate and polyethylene glycol and discussed the quality of the ensuing lignin-based polyol [15,16]. More recently, we have studied the use of different catalysts on the oxyalkylation of Lignoboost^TM^ kraft lignin with propylene carbonate and concluded that DBU was the best catalyst for this type of lignin. The DS obtained was 0.78 and the chain length was 1.44 [17].

As is well known, the production of PU is very dependent on the characteristics of the polyol; hence, in the polyol industry, several analyses are used to characterize product performance such as hydroxyl number (I_OH_), viscosity, functionality and molecular weight. The required values depend on the final product, such as the production of foams, films or adhesives [18]. For instance, the I_OH_ is a key parameter in the formulation of PUs, since it is defined as the number of OH groups available for the reaction with the isocyanate. Consequently, the balance between the NCO and OH groups affects the extent of the PU crosslinked networks and subsequently their properties. The range of I_OH_ values required to produce PU is quite wide. In general, conventional polyether polyols for PU rigid foams have a range of I_OH_ values between 300 and 800 mg KOH/g [18]. In turn, I_OH_ values found in the literature for lignin-based polyols obtained by oxyalkylation range from 164 to 569 mg KOH/g. Moreover, such lignin-based polyols with a higher number of hydroxyl groups also have higher viscosity [9,16,19].

The viscosity of polyols is another important parameter as it has a major impact on the mixing of PU formulations. Indeed, obtaining low viscosity of lignin-based polyols can be a challenging task and this is a limiting feature if they are to be used in PU formulations because they are difficult to stir. Both I_OH_ values and viscosity can be changed via liquefaction as mentioned above [15,16,17]; regarding the oxyalkylation of lignin with cyclic carbonates to yield polyols suitable for the production of PU, there is still a lack of full understanding of the oxyalkylation optimization process of kraft lignin to tune the characteristics of the ensuing polyols. Thus, the use of tools such as the design of experiment (DoE) is necessary. The DoE is a powerful data collection and analysis tool, which enables finding the most important features of the system to be optimized. Within the different DoE approaches, full factorial experiment combined with response surface methodology (RSM) is an effective methodology to optimize processes with multi-response.

The main goal of this work was to optimize the oxyalkylation of Lignoboost^TM^ kraft lignin with PC using RSM. The oxyalkylation process variables were temperature, time and quantity of PC whilst keeping the amount of catalyst (DBU) fixed. The targeted quality parameters of lignin-based polyols (I_OH_ and viscosity) were close to those of conventional polyols typically used in the synthesis of rigid PU foams and PU adhesives. In addition, RSM was employed to minimize the amount of PC involved in the synthesis whilst ensuring the quality of lignin-based polyol.

## 2. Materials and Methods

### 2.1. Materials

Lignoboost™ kraft lignin (KL) obtained from kraft pulping of Eucalyptus globulus was kindly supplied by a Portuguese pulp and paper mill (Cacia, Portugal). Propylene carbonate (PC) was purchased from Acros Organics Company (Geel, Belgium) and was used without any further purification; 1,8-diazabicyclo[5.4.0]undec-7-ene (DBU) was supplied by Aldrich Chemical Company (St. Louis, MO, USA) and used as received.

### 2.2. Oxyalkylation of Kraft Lignin

A desired amount of KL (1 equivalent molar), PC (4, 7, and 10 eq molar) and DBU (0.1 eq. molar) were placed in a 50 mL two neck round-bottom flask, equipped with a reflux condenser, mechanical stirrer and thermometer. The reaction mixture was stirred at 300 rpm for 1.5, 2.5 and 3.5 h at 170, 180 and 200 °C, respectively, under N_2_ atmosphere. These parameters were selected based on analogous studies reported in the literature regarding the oxyalkylation reaction using cyclic carbonates, especially with phenolic compounds such as lignin and tannin [11,13,14,16,20,21,22]. Prior to oxyalkylation, the lignin samples were vacuum dried in an oven at 30 °C to eliminate moisture. Typically, 5.00 g of lignin (equivalent to 6.96 mmol of total OH groups, 1 eq. molar), 6 mL of PC (69.6 mmol, 10 eq. molar) and catalyst DBU (0.696 mmol, 0.1 eq. molar) were sequentially added into a 50 mL two neck round-bottom flask. The mixture was stirred at 170 °C for 1.5 h. Notice should be made that one mole of lignin corresponds to one phenylpropane unit (PPU) C_9_H_6.76_O_2.95_S_0.18_(OCH_3_)_1.41_ (Mr = 211.4 g/mol) as determined in previous work [17]. This means that 1 g of lignin corresponds to 1/211.4 mol of lignin or 6.96 mmol of OH groups, taking into account that OH total = 1.47 per one PPU. The reaction product was a viscous polyol after cooling to room temperature. This crude product, which consists of PC-modified lignin and PC homoligomer, was dried in a vacuum oven at 30 °C to constant weight, to yield the lignin-based polyol. For comparison, samples of kraft lignin and of the lignin-based polyol are presented in Figure 1.

### 2.3. Design of Experiments (DoE)

The flowchart of the DoE study is shown in Figure 2. The first step of DoE consisted in selecting the process variables. Hence, based on data available in the literature and the experience of the group temperature, time and amount of PC were chosen as process variables (or independent variables) to assess the effect on the characteristics of the polyol (I_OH_ and viscosity). Regarding the amount of PC, the lowest limit considered was the minimum amount required to obtain a product that was viscous but not pasty so that it could be stirred which corresponded to 4 eq. molar of PC. The type of design chosen was the full factorial with three levels and three variables (3^3^) as shown in Table 1.

The statistical analysis was carried out using the analysis of variance method (ANOVA) using the JMP^®^ software (version trial, SAS Institute Inc., Cary, NC, USA). The general form of the second-order polynomial equation for this experimental design with three process variables and three levels is given by Equation (1). These three levels yield a model with curvature known as response surface.
(1)$$Y_{i}=\beta_{0\;}+\beta_{1}X_{1}+\beta_{2}X_{2}+\beta_{3}X_{3\;}+\beta_{12}X_{1}X_{2}+\beta_{13}X_{1}X_{3}\;+\beta_{23}X_{2}X_{3}+\beta_{11}X_{1}^{2}\;\beta_{22}X_{2}^{2}\;\beta_{33}X_{3}^{2}\;$$
where *Yi* is the expected response (dependent variable) and *X*_1_, *X*_2_ and *X*_3_ are the process variables (independent variables) temperature, time and PC amount, respectively; *β*_0_ is the constant coefficient and *β*_1_, *β*_2_ and *β*_3_ are the linear coefficients (also called main effects); *β*_12_, *β*_13_ and *β*_23_ are the cross-product coefficients (interaction between the process variables) and *β*_11_, *β*_22_ and *β*_33_ are the quadratic coefficients used to model the curvature of the response surface.

### 2.4. Lignin-Based Polyol and Lignin Characterization

The hydroxyl number (I_OH_) of the polyols was determined by potentiometric titration of residual acetic acid after esterification of the free OH groups [23]. The procedure was as follows: Approximately 20 mg of sample was weighed into a screw cap tube. About 0.1 mL of acetylation mixture was added, which had been prepared just before the analyses by mixing of 4.7 mL of acetic anhydride (Ac_2_O) and 4 mL of pyridine. The tube content was homogenized and kept for 24 h in an oven at 50 °C. After cooling to room temperature, the mixture was transferred quantitatively with 10 mL of acetone to a 100 mL beaker. An equal amount of distilled water was added, and the mixture was titrated using 0.1 N LiOH. The average value of three replicates was obtained.

The percentage of total OH groups was calculated using Equation (2). The conversion of % OH groups to number of milligrams of KOH required to neutralize one gram of sample was calculated using Equation (3).
(2)$$\mathrm{OH}\left(\%\right)=\frac{\left[(m_{s}\cdot V_{b}/m_{b})-V\right]\cdot f\cdot 1.7\times 100}{W}\;$$
I_OH_ (mg KOH/g) = 33 × (% OH)(3)
where *V* is the volume of LiOH solution required for the titration of the sample (mL); *V_b_* is the volume of LiOH solution required for the titration of the blank (mL); *m_s_* is the acetylating mixture of sample (mg); *m_b_* is the blank (acetic anhydride and pyridine) in mg; *f* is the standardized titer of LiOH solution; *W* is the weight of the sample (mg); 1.7 is the mass, in mg, of hydroxyl groups equivalent to 1 mL of 0.1 M LiOH.

The viscosity of lignin-based polyol was determined at 25 °C using a Kinexus lab+ rotational rheometer, with a 4° cone geometry of 40 mm diameter and a gap of 150 µm. The viscosities were measured in rotational mode using a shear rate sweep between 0.1 and 100.0 s^−1^, while measuring 5 points per decade.

The Fourier transform infrared (FTIR) spectra were recorded on an FTIR System Spectrum BX (PerkinElmer) coupled with a universal ATR sampling accessory, in reflectance mode (%) from 4000 to 600 cm^−1^ by accumulating 64 scans with a resolution of 4 cm^−1^.

The size exclusion chromatography (SEC) analysis was performed using the PL-GPC 220 system (Agilent) equipped with two columns Agilent PL gel MIXED-D, 7.5 × 300 mm, 5 µm (in series) protected by a PL gel MIXED precolumn, 7.5 × 50 mm, 5 µm. The columns, injector system and the detector (RI) were maintained at 70 °C during the analysis. The eluent (0.5% *w*/*v* LiCl in DMF) was pumped at a flow rate of 0.9 mL min^−1^. The columns were calibrated using lignin model compounds and lignin samples (Mp = 950–3200 Da) previously characterized by electrospray ionization mass spectrometry (ESI-MS).

The quantitative ^1^H NMR spectra were e recorded on a BRUKER AVANCE III 300 spectrometer operating at 300.13 MHz (298 K) with the following acquisition parameters: 12.2 µs pulse width (90°), 2 s relaxation delay and 300 scans. Samples of acetylated lignin and acetylated lignin-based polyol (15 mg) were dissolved in 0.5 mL deuterated chloroform (CDCl_3_) prior to analysis.

The water content of the crude polyols was obtained by using Karl–Fischer titration. The samples were diluted with methanol and titrated with Fischer reagent.

## 3. Results and Discussion

In our previous studies regarding the oxyalkylation of lignin using PC it was demonstrated that the synthetic path and thus the product depended on the type of catalyst as depicted in Figure 3. The best results were obtained using DBU as catalyst which promoted the preferential deprotonation of phenolic OH groups of lignin towards reactions with PC [17]. The goal of the current study was to optimize the synthesis of lignin-based polyol from oxyalkylation of kraft lignin with PC. For that purpose, the amount of DBU was fixed and the effect of other process variables (temperature, time and quantify of PC) on the quality of lignin-based polyol (in terms of I_OH_ and viscosity) was evaluated using RSM. Finally, the optimized polyols were characterized and compared with that obtained from our previous studies as discussed in Section 4.

### 3.1. The Mathematical Model and Its Evaluation

Table 2 shows the operating conditions considered for the 3^3^ full factorial design, corresponding to a total of 27 experiments, as well as the experimental data and model-predicted values obtained for the responses, i.e., I_OH_ and viscosity. These experimental data were used to generate the models for the responses. Normally, RSM uses other types of design with three levels, for example, central composite design (CCD) and Box Behnken design. However, in this work a full factorial design 3^3^ was used to collect the data, and the Response Surface effect attribute (RS) was selected in the JMP software package (version trial) for each independent variable (main effect) to build the model for the responses. This choice was because the 3^3^ design has the middle level of process variables and the curvature can be determined. In addition, the parameters of the second-order polynomial equations of responses (Equations (4) and (5)) were estimated based on the original values of the process variables, i.e., uncoded levels.

Multiple linear regression with the least square fitting was used to find the best line or curve for the responses. For viscosity data, the fitted line plot shown in Figure 4a indicates that the experimental data fit reasonably well with the model, and the correlation coefficient (R-squared) is 0.91. However, a closer analysis reveals that the predicted values of viscosity are negative. Furthermore, the viscosity residual plot (Figure 4b) shows a distinct deviation from the usual linear model. This is a clear indication of the absence of linearity between the response and the predicted values. Indeed, it is well known that the viscosity of many materials such as polymers does not follow linear behavior but a logarithmic one [24,25]. Hence, the log values of viscosity were used instead and as can be observed in Table 2 and Figure 5b, both the experimental and predicted values follow a linear trend.

The fitted regression model equations for the I_OH_ (*Y_IOH_*) and viscosity (*Y_Log_*_10[*viscosity*]_) are described by Equations (4) and (5), respectively.
(4)$$\begin{array}{ll} Y_{IOH\;}=436.48 & +76.72Temp+131.04Time-38.19PC-39.62Temp\cdot Time\\ & -0.71Temp\cdot PC-6.83Time\cdot PC-119.83Temp^{2}\\ & +84.55Time^{2}-20.77PC^{2} \end{array}$$
(5)$$\begin{array}{ll} Y_{Log10\left[visco\right]}= & 0.723+0.304Temp+0.206Time-0.693PC\\ & -0.214Temp\cdot Time-0.095Temp\cdot PC+0.014Time\cdot PC\\ & -0.264Temp^{2}-0.047Time^{2}+0.231PC^{2} \end{array}$$

Figure 5 shows the good consistency between the experimental values and the values predicted by the models for both I_OH_ and viscosity. The R-squared for these models were 0.92 and 0.98, respectively, indicating the goodness of the fit of the regression models.

Additionally, the residual analysis, presented in Figure 6, was performed to verify if the assumptions subjacent of the analysis of variation are valid, that is, if the errors are independent and normally distributed with constant variance, as expected in linear regression. The plot of residuals versus predicted values for I_OH_ and viscosity indicated that the residuals appear to be randomly distributed around the line that corresponds to the zero-residual value, with constant dispersion, and no tendency being observed. This indicates that the variance is constant and that there is a relationship between the variables; thus, the fitted regression model is suitable.

The adequacy of the fitted regression model for I_OH_ and viscosity was verified by ANOVA (Analysis of Variance) using the Fisher F-test and its associated probability (*p* value) for a 95% confidence level.

The results of ANOVA are shown in Table 3 and, as can be observed, the *p* values are <0.05 indicating that the models are significant, in other words, the process variables and their interactions have an effect on the responses.

The significance of the main effect and the interaction of process variables of fitted regression models were also evaluated using the Pareto chart (see Figure 7). In this chart, the model effects for both responses, I_OH_ and viscosity, are organized by ascending *p* values for an interval of confidence of 95%. Notice should be made that logWorth is a *p* value transformation, defined as −log_10_ (*p* value); hence, a value that exceeds 2 is significant at the 0.01 level which is illustrated by the blue line in the chart.

As can be observed in Figure 6, the linear effects are the most influential, especially the amount of PC. Next, the interaction effect Temp * Time also shows some significance. However, the interaction effects Temp * PC and Time * PC are not statistically significant in an interval of confidence of 99%, but the interaction effect Temp * PC is significant considering an interval of confidence of 95%. Finally, the quadratic effects (responsible to fit curves to the data in 3D plots, such as those shown in Figure 8 and Figure 9) also have significance, but are not as relevant as those of the linear effects. In short, most of the interaction terms have some level of significance, indicating that the relationship between one variable and the response depends on the other variables. In fact, only the interaction effect Time * PC does not significantly affect the response.

### 3.2. The Effect of Process Variables on the Hydroxyl Number and Viscosity of Lignin-Based Polyol

The three-dimensional response surface plots (3D plots) were generated using the model Equations (4) and (5), with the purpose to understand the interaction effect of the process variables (temperature, time and PC amount) on the responses (I_OH_ and viscosity).

#### 3.2.1. Hydroxyl Number (I_OH_)

Prior to discussing the plots, it must be noted that the models have three variables, and that one variable was kept constant at the medium level for each plot.

The interaction effect Temp * Time is illustrated in Figure 8a. As already indicated by the Pareto chart, this type of interaction seems to have a high influence on the values of I_OH_. Indeed, longer times (>3.0 h) and higher temperature lead to higher values of I_OH_. Since the oxyalkylation reaction occurs via opening the PC ring by the attack of phenoxide to the alkylene carbon of PC, leading to chain extension [17]; it seems that the increase of time and temperature of oxyalkylation reaction allows better accessibility of phenolic groups of lignin. This can be associated with the fact that as the phenolic OH are modified the intricate structure of lignin loosens up, making other phenolic OH groups more available as well as facilitating diffusion of PC and catalyst to more hindered sites. Moreover, it should be kept in mind that, as shown in our previous work, during the oxyalkylation of lignin with PC, there is a small conversion of PC to homopolymer and this fraction of homopolymer contributes to the increase of the I_OH_ [17]. Interestingly, the same trend regarding the effect of the reaction time and temperature was observed in the oxyalkylation lignin with ethylene carbonate and polyethylene glycol reported by Xuefeng et al. [16]. In contrast, low values of I_OH_ are obtained when the process is carried out at lower temperatures and shorter times.

As regards the interaction effects involving the amount of PC, the Pareto chart shows that the interaction effects, Time * PC and Temp * PC, do not have significant impact on the responses. This means that the effect of time and temperature on the responses does not depend on the amount of PC used. Furthermore, the effect of process variables on the curvature of the surface, illustrated in Figure 8b,c, confirms that the effect of the amount of PC is rather independent of the other process variables. While from the interaction effect Time * PC, shown in Figure 8b, it can be observed that for longer times and smaller amounts of PC the I_OH_ increases, this effect is mostly associated with time, suggesting that the process is diffusion controlled. In turn, the interaction effect Temp * PC illustrated in Figure 8c reveals that the effect of the amount of PC is not significant, yet this may be due to the fact that the range of amount of PC used was small. However, when the temperature increases and the amount of PC decreases, the I_OH_ increases. This may be associated with the fact that more oligomers are formed as the smaller amount of PC limits diffusion of the reagents towards the hindered phenolic groups of lignin due to the higher viscosity of the reaction mixture as discussed next.

#### 3.2.2. Viscosity

The interaction effects between the process variables on the viscosity are depicted in Figure 9a–c.

The interaction effect Temp * Time in Figure 9a shows that the general tendency is an increase in viscosity with the increase of reaction temperature and time, where the effect of temperature is more pronounced than that of the reaction time. It is well known that the viscosity of a polyol increases with increasing reaction temperature [18]. Yet, few studies in the literature have related directly the variation (increase) of the viscosity during the oxyalkylation of lignin using cyclic carbonate with temperature [16]. Furthermore, other studies report the effect of temperature on the grafting of the alkyl units on lignin, which probably lead to higher viscosity even though this was not stated [26,27,28].

Figure 9b,c illustrate that the viscosity is highly influenced by the amount of PC. In general, the viscosity increases with smaller amounts of PC because it acts both as solvent and as reagent. In turn, higher amounts of PC lead to less viscous lignin-based polyol. In fact, the significance of the amount of PC was already highlighted by the Pareto chart. Notice should also be made that when the amount of PC was reduced from 10 eq. molar to 4 eq. molar the lignin content in the polyol increased from 12.5 wt % to 26.6 wt %. This tendency was also observed in the oxyalkylation of lignin with ethylene carbonate and propylene oxide [9,16]. As regards the interaction effect Temp * PC this seems to have some effect when the process is carried out using a very small amount of PC and high temperatures leading to an increase in viscosity. However, the interaction effect Time * PC is basically insignificant.

### 3.3. Validation of the Fitted Regression Models

To validate the conditions predicted by the models, three additional tests were carried out using different conditions. The process conditions used and results obtained are presented in Table 4. The adequacy of the model equations for predicting response values was verified by comparing the observed and predicted values for the different conditions, using low, middle and high amounts of PC. The observed values were coherent with the predicted values since the latter were within the confidence interval for the model. This leads to the conclusion that this type of model can be set up after performing a statistically satisfactory minimal number of experiments to estimate the response.

### 3.4. Optimization of the Process

Bio-based polyols are becoming a commercial reality, and lignin-based polyols could become a commercial product as well. To achieve that, process optimization, even on a laboratory scale, is extremely relevant to obtain a product that is economically viable and competitive with conventional polyols.

Usually, process optimization involves multi-responses, with the desirability function being the most widely used method to allow the simultaneous optimization of several responses. This method consists in converting each response *y_i_* into an individual desirability function *d_i_* that is then aggregated into a composite function (*D*) [29,30]. This function is usually a geometric or arithmetic mean, which will be maximized, minimized or even matched to targets, according to the objective of the optimization. The values of desirability range between 0 and 1, where zero indicates unacceptable quality and 1 is associated with the best response. The generic form of the mathematical relationship between responses and the desirability function is given by Equation (6).
(6)$$D=(d_{1}.d_{2}\ldots d_{m}){\;}^{1/m}$$
where *m* is the number of responses. In the present work, the desirability function is associated by the prediction profiler (a tool of JMP^®^ software, version trial) with two responses, I_OH_ and viscosity. Before optimizing the process, the goal for each response was defined based on information available in the literature and data from the conventional polyol industry. The I_OH_ and viscosity values of polyols found in the market depend on the final application such as rigid or flexible foams, adhesives, films, etc. In this work, the main goal is to produce lignin-based polyol for the production of rigid foams and adhesives for wood gluing. Following those distinct objectives, the optimization should be different for each product.

#### 3.4.1. Rigid Foams

Usually, polyether polyols used to produce rigid foams have a wide range of I_OH_ values of 300–800 mg KOH/g, and viscosities in the range of 0.3–30 Pa.s, which varies with the molecular weight and functionality [18,31]. Based on the I_OH_ and viscosity values of polyols in the market and the values of lignin-based polyol found in the literature, the desirability function was run for the two responses to match a target value. The assigned target for I_OH_ (mg KOH/g) was: 90 (low), 380 (middle) and 600 (high). For viscosity (Pa·s) it was 0.00 (low), 5.0 (middle) and 30 (high).

The prediction profiler presented in Figure 10 shows how the response computation changes with the input variables over the range established and illustrates the response surface in the two-dimensional plot. Moreover, the profiler shows that the operating process conditions to obtain the predicted responses with high desirability are in the following intervals: Temperature 180–185 °C; reaction time, 1.8–2.5 h; and amount of PC, 5.0–7.0 eq. molar of PC. In other words, these are the ranges of operating conditions for the lignin oxyalkylation process considered ideal to produce a polyol to be used in the formulation of rigid foams. Of significant relevance is the fact that this allowed increasing the lignin content in the polyol from 12.5 to 25% (*w*/*w*), i.e., the initial amount of PC (10 eq. molar) used to produce polyol can be reduced to 5 eq. molar of PC whilst ensuring that the quality parameters are kept.

#### 3.4.2. Adhesives

Polyols are also used in the formulation of other polyurethane products such as coatings, adhesives, sealants and elastomers, where the specification for I_OH_ and viscosity is quite diverse. In general, they have I_OH_ and viscosity values lower than the polyols used to produce rigid foams [18]. PU adhesives are suitable for various materials such as metal, wood, plastic, etc. For wood adhesives, in particular, the adhesive penetration can be strongly affected by the viscosity of the PU, with high viscosities yielding lower penetration [32,33]. The viscosity of the adhesive must be such that it penetrates into the voids of the wood but not too fluid to prevent dispersing out from the substrate. Based on studies using lignin-based polyols and conventional polyols to produce PU adhesives, the range of I_OH_ and viscosity to optimize the operating conditions was selected [31,32,34,35]. The assigned targets for I_OH_ (mg KOH/g) were 90 (low), 235 (middle) and 380 (high) and for viscosity (Pa.S) were 0.00 (low), 1.0 (middle) and 10 (high).

Figure 11 shows that the operating conditions to obtain predicted responses with high desirability to produce polyols for polyurethanes adhesives are within the following intervals: Temperature 170–175 °C; reaction time 1.5–2.0 h; and amount of PC 5.5–7.5 eq. molar of PC. These operating conditions also allowed to increase the lignin content in the polyol from 12.5 to 22.5% (*w*/*w*), increasing the renewable content of this type of adhesive.

## 4. Characterization of Lignin-Based Polyols from Optimized Conditions

The chemical structure of the non-optimized and optimized polyols (designated as crude polyols) was evaluated by FTIR-ATR, NMR analysis and the molecular weight by SEC. Under non-optimized oxyalkylation conditions published previously [17], the lignin-to-PC molar ratio was 10 eq. molar of PC, the reaction was carried out at 170 °C for 2.5 h and the final product was purified. According to optimization results using RSM, for rigid PU foam application, the crude polyol was obtained using the lignin-to-PC molar ratio 5.5, 180 °C and 2.0 h of reaction, while the crude polyol for PU adhesive application was synthetized using lignin-to-PC molar ratio 7.5, 170 °C and 2.0 h of reaction.

The successfulness of oxyalkylation of lignin was confirmed by FTIR-ATR (Figure 12), showing a decrease and shift of a broad peak at 3600–3200 cm^−1^ due to the reduction of hydrogen bonding of reacted phenolic groups and the appearance of the new peak at 1820–1720 cm^−1^ of C=O stretching in the carbonate structures resulting from the reaction of PC. The increase in band intensity around 1050 cm^−1^ can be attributed to the C-O stretching vibration of newly formed alcohol groups. These spectrum features are coherent with the proposed scheme of lignin oxyalkylation reactions shown in Figure 3.

The chemical and physical characteristics of polyols are shown in Table 5. Optimized crude polyols have lower weight-average molecular weight (M_w_) than the non-optimized one due to the presence of low molecular weight products such as oligomeric glycols. The crude polyol for PU adhesive applications showed lower M_w_ (1700 Da), lower viscosity and higher content of water. Probably, the PC underwent hydrolysis leading to the formation of more homopolymer, as this polyol was synthesized using a higher proportion of PC than the polyol optimized for the production of foams. It is important to mention that the presence of low amounts of homopolymer in the final product in terms of polyurethane synthesis is not a drawback, as both polyol and homopolymer react with the isocyanate.

Quantitative differences between lignin and lignin-based polyols were estimated by ^1^H NMR spectroscopy (Figure 13). The oxyalkylation of kraft lignin into liquid polyols is confirmed by the disappearance of the methyl protons in the acetylated phenolic OH (2.1–2.2 ppm) and the appearance of new methyl groups in isopropyl moieties (1.0–1.2 ppm) for all acetylated samples. The confirmation of homopolymer presence in the optimized crude polyols is revealed by the protons from oxygenated CH, CH_2_ (3.6–5.2 ppm) moieties in oligomeric glycols, which was not observed in the non-optimized purified polyol.

The degree of substitution (DS) of polyols was calculated based on the amounts of reacted original phenolic OH in kraft lignin [17]. In turn, the number of estimated oxypropyl units per one phenyl propane lignin unit (PPU) of crude optimized polyols was higher than that of non-optimized polyol (Table 5). This can be explained by the presence of oligomeric glycols not necessarily grafted onto lignin that were not removed during the purification step as for the non-optimized polyol. In addition, the presence of a noticeable amount of homopolymer in crude optimized polyols was confirmed by strong resonances at 1.9–2.0 ppm belonging to methyl protons in acetylated aliphatic OH [17].

## 5. Conclusions

The application of DoE using the full factorial and RSM methodology in this study enabled extracting information about how the conditions of the oxyalkylation process using PC affect the quality of the lignin-based polyol. The response surface generated from second-order equations allowed to verify that the variables PC amount, temperature, and time have great impact on the responses. Moreover, this methodology also allowed to find the optimal range of operating conditions to obtain lignin-based polyol with desired values of I_OH_ and viscosity. Furthermore, from the desirability function, the operating conditions of the oxyalkylation process were optimized, which allowed to reduce the amount of PC necessary to obtain lignin-based polyol that can be used in formulations of rigid foams and adhesives for wood bonding. The chemical and physical characteristics of the optimized polyols obtained confirmed the main structural features previously reported being the differences associated with the presence of homopolymer. This optimization showed that, depending on temperature and time of reaction, the quantify of PC can be reduced in circa 50% and the lignin content in the polyol can be increased from 12.5% up to 25% (*w*/*w*). Thus, the amount of renewable content has been significantly increased. Yet, different types of lignins, as well as distinct applications, may naturally require further fine tuning of the process variables.

## Figures and Tables

**Figure 1 materials-15-01925-f001:**
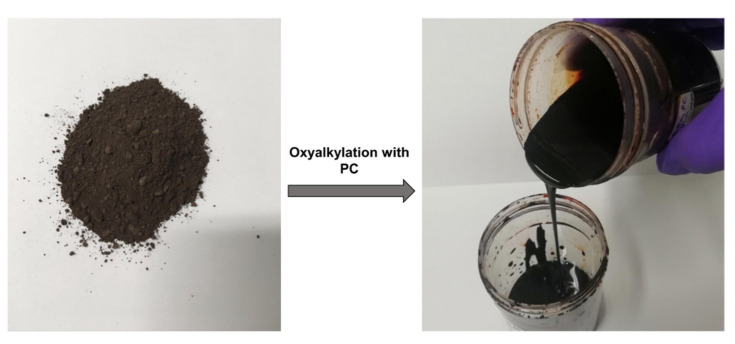
Kraft lignin and lignin-based polyol obtained by oxyalkylation using PC.

**Figure 2 materials-15-01925-f002:**
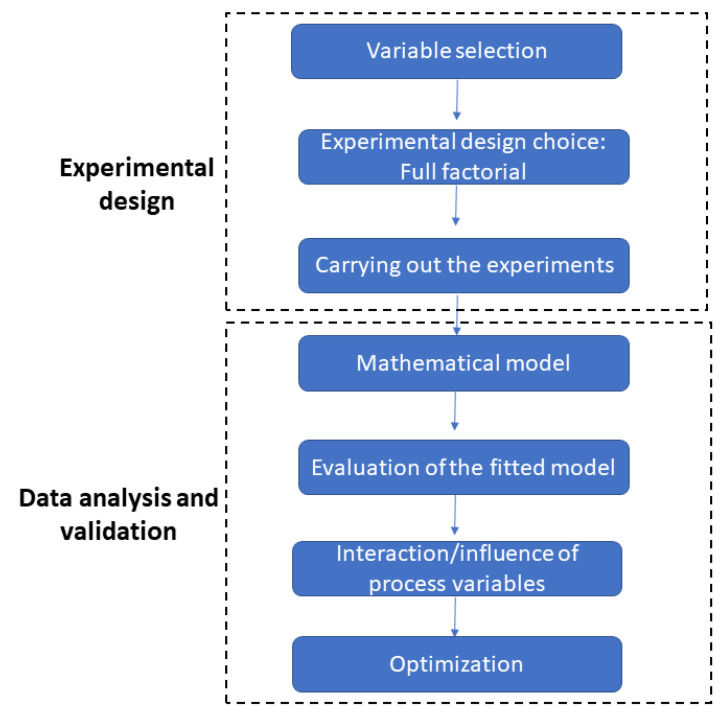
The flowchart of the DoE study.

**Figure 3 materials-15-01925-f003:**
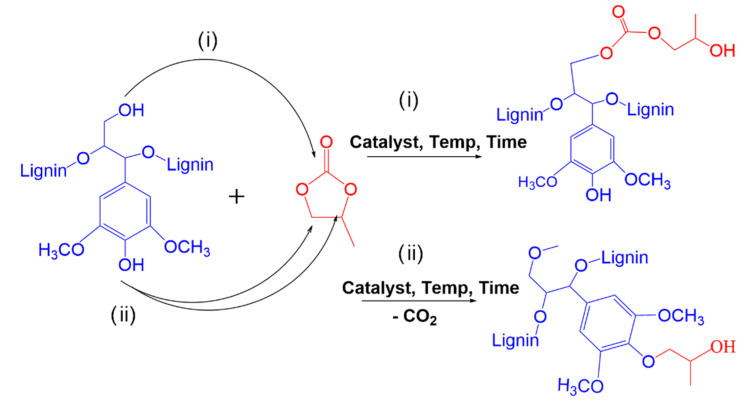
Main synthetic paths involved in lignin oxyalkylation with PC (i) PC can react with aliphatic OH, (ii) PC can react phenolic OH [17].

**Figure 4 materials-15-01925-f004:**
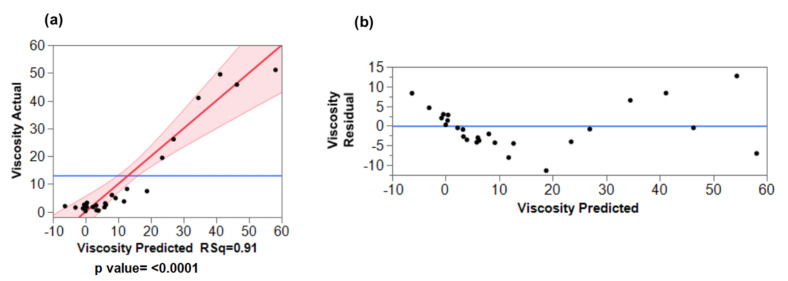
(**a**) Plots of actual versus predicted values of viscosity, (**b**) Plot of viscosity residual versus predicted values.

**Figure 5 materials-15-01925-f005:**
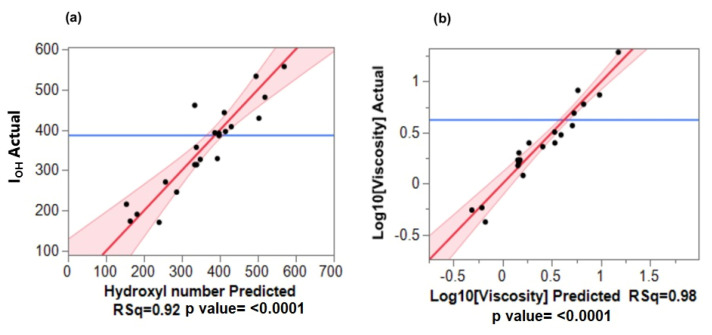
(**a**) Plots of actual versus predicted values of I_OH_, (**b**) Plots of actual versus predicted values of viscosity.

**Figure 6 materials-15-01925-f006:**
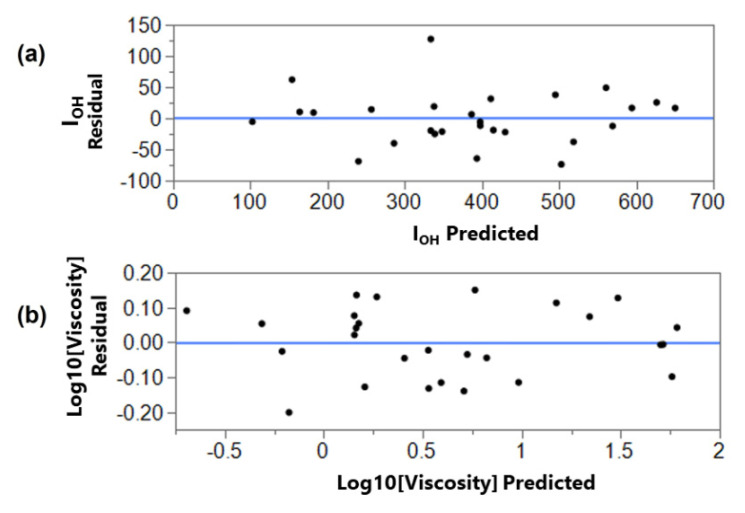
Plots of Residual versus predicted values for (**a**) I_OH_, (**b**) Viscosity.

**Figure 7 materials-15-01925-f007:**
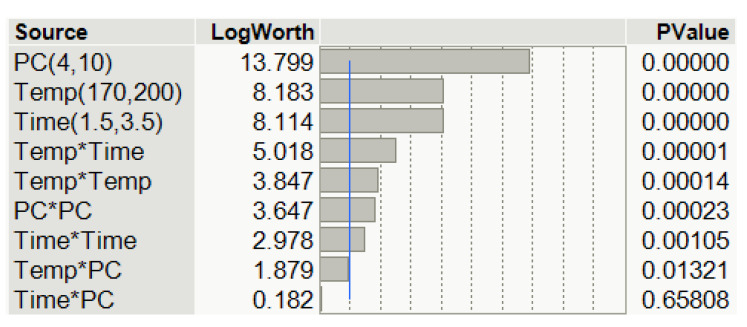
Pareto chart of the main effects obtained from the three levels (3^3^ ) full factorial design.

**Figure 8 materials-15-01925-f008:**
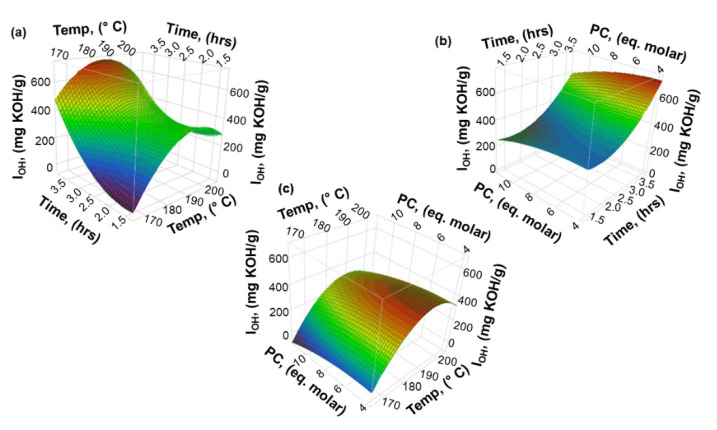
Three-dimensional surface plots for the hydroxyl number (I_OH_) of lignin-based polyol. (**a**) The interaction effectTemp. and Time on the hydroxyl number, (**b**) The interaction effect Time and PC on the hydroxyl number, (**c**) The interaction effect Temp. and PC on the hydroxyl number.

**Figure 9 materials-15-01925-f009:**
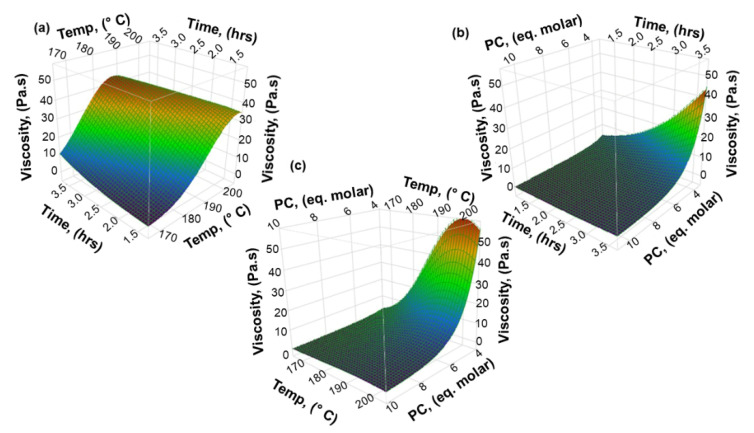
Three-dimensional surface plots for the viscosity of lignin-based polyol: (**a**) The interaction Temp. andTime effect on the viscosity, (**b**) The interaction effect Time and PC on the viscosity, (**c**) The interaction effect Temp. and PC on the viscosity.

**Figure 10 materials-15-01925-f010:**
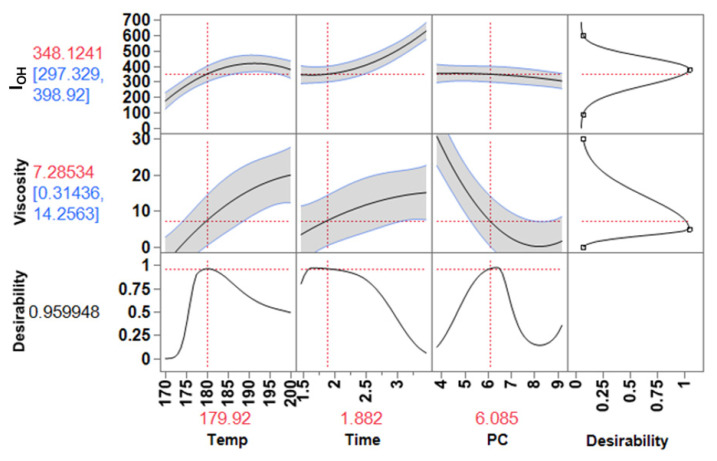
Prediction profiler and desired conditions for required I_OH_ and viscosity of lignin-based polyol to produce rigid foams.

**Figure 11 materials-15-01925-f011:**
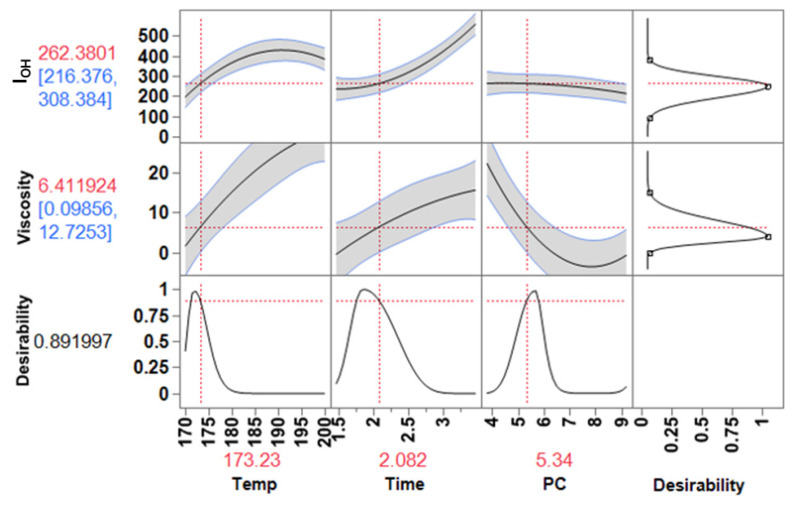
Prediction profiler and desired conditions for required I_OH_ and viscosity of lignin-based polyol to produce polyurethane adhesives.

**Figure 12 materials-15-01925-f012:**
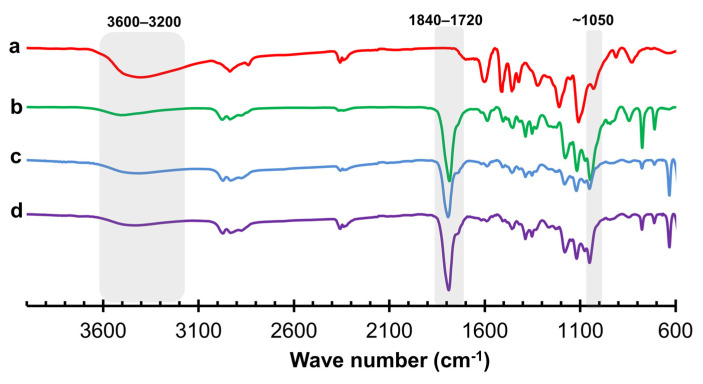
FTIR normalized spectra of (**a**) kraft lignin, (**b**) Non-optimized polyol, (**c**) Crude lignin-based polyol for rigid PU foam and (**d**) Crude lignin-based polyol for PU adhesives.

**Figure 13 materials-15-01925-f013:**
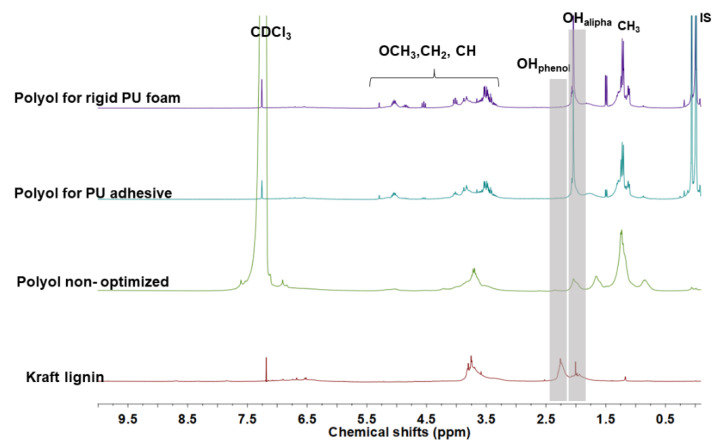
Quantitative proton (^1^H) spectra NMR of acetylated kraft lignin, non-optimized polyol and crude polyols for rigid PU foam and PU adhesive.

**Table 1 materials-15-01925-t001:** Full-factorial design of experiments with three process variables and three levels (3^3^) for the oxyalkylation of kraft lignin with PC.

Process Variables	Levels		
	1	2	3
Temperature (Temp), °C	170	180	200
Time (Time), hours	1.5	2.5	3.5
Quantify of PC (PC), eq. molar	4	7	10
**Responses**			
Hydroxyl number (I_OH_), mg KOH/g			
Viscosity, Pa·s			

1: Low level; 2: Intermediate level; 3: High level.

**Table 2 materials-15-01925-t002:** Design matrix and the results of experiments.

Run	Process Variables				Responses		
	Temp, °C	Time, Hour	PC, eq. Molar	Experimental I_OH_, mg KOH/g (Mean ± * SD)	Predicted I_OH_, mg KOH/g	Experimental Viscosity at 25 °C, Pa·s (Mean ± * SD)	Predicted Viscosity, Pa·s
1	170	1.5	4	174 ± 3.6	163	2.5 ± 0.49	3.4
2	170	1.5	7	216 ± 8.8	154	0.55 ± 0.13	0.49
3	170	1.5	10	97 ± 8.0	102	0.25 ± 0.10	0.20
4	170	2.5	4	271 ± 19.1	256	7.4 ± 1.2	9.6
5	170	2.5	7	171± 3.6	240	1.5 ± 0.35	1.4
6	170	2.5	10	191 ± 9.3	182	0.58 ± 0.17	0.61
7	170	3.5	4	533 ± 16.5	519	26.1 ± 5.2	22.0
8	170	3.5	7	481 ± 15.3	495	3.2 ± 0.98	3.3
9	170	3.5	10	408 ± 18.9	430	1.7 ± 0.40	1.5
10	180	1.5	4	327 ± 16.6	348	19.4 ± 5.4	15.0
11	180	1.5	7	357 ± 9.4	337	2.5 ± 0.61	1.9
12	180	1.5	10	246 ± 13.1	286	0.42 ± 0.13	0.66
13	180	2.5	4	396± 48.4	415	41.0 ± 5.3	30.6
14	180	2.5	7	392 ± 36.2	398	3.0 ± 0.46	3.9
15	180	2.5	10	314 ± 10.6	339	1.6 ± 0.41	1.5
16	180	3.5	4	667 ± 38.7	650	49.5 ± 6.0	50.3
17	180	3.5	7	652 ± 11.3	626	6.0 ± 1.9	6.6
18	180	3.5	10	610 ± 10.2	560	2.3 ± 0.43	2.6
19	200	1.5	4	386 ± 27.3	398	45.8 ± 4.8	57.4
20	200	1.5	7	393 ± 26.6	386	4.9 ± 1.4	5.3
21	200	1.5	10	314 ± 26.9	334	1.7 ± 0.22	1.4
22	200	2.5	4	443 ± 22.2	411	67.1 ± 6.4	60.8
23	200	2.5	7	329 ± 15.1	393	8.2 ± 2.2	5.8
24	200	2.5	10	461 ± 36.8	334	1.2 ± 0.36	1.6
25	200	3.5	4	611 ± 20.6	594	51.1 ± 5.4	51.8
26	200	3.5	7	557 ± 22.0	569	3.7 ± 0.53	5.1
27	200	3.5	10	429 ± 35.5	503	2.0 ± 0.44	1.5

* Standard deviation (SD).

**Table 3 materials-15-01925-t003:** ANOVA results for the regression models for I_OH_ and viscosity.

Source	Responses					
	I_OH_			Viscosity		
	DF	SS	MS	DF	SS	MS
Model	9	567,217.54	63,024.2	9	12.13	1.34
Error	17	46,945.86	2761.5	17	0.252	0.014
Total	26	614,163.41	-	26	12.38	-
F ratio	22.82			90.72		
*p* value	<0.0001			<0.0001		
R square	0.923			0.979		
R-square adjusted	0.883			0.9688		
Mean of response	386.14			0.62		

DF: Degree of freedom; SS: Sum of square; MS: Mean of square.

**Table 4 materials-15-01925-t004:** Validation of the fitted model for I_OH_ and viscosity.

Run	Variables				Responses		
	Temp, °C	Time, Hour	PC, eq. Molar	Predicted I_OH_, mg KOH/g (95% Confidence Interval)	Observed I_OH_, mg KOH/g (Mean ± * SD)	Predicted Viscosity, Pa·s (95% Interval Confidence)	Observed Viscosity at 25 °C, Pa·s (Mean ± * SD)
1	170	2.5	10	182 (120–242)	237 ± 13.1	0.62 (0.44–0.85)	0.51
2	180	2.0	5.13	372 (324–420)	411 ± 12.9	9.90 (7.5–12.6)	7.70
3	200	1.5	7.0	390 (322–451)	437 ± 29.1	5.50 (3.8–7.5)	5.40

* Standard deviation (SD).

**Table 5 materials-15-01925-t005:** Chemical and physical characteristics of lignin and lignin-based polyols.

Characteristics	Kraft Lignin *	Polyol Non- Optimized *	Crude Polyol for Rigid PU Foam	Crude Polyol for PU Adhesive
Water content, wt %	-	-	0.16	0.39
I_OH_, mg KOH/g	311	198	257	225
Viscosity, Pa·s	-	0.58	5.3	0.56
M_w_, Da	1415	1970	1760	1700
Polydispersity	1.09	1.11	1.09	1.08
Oxypropyl units	-	1.44	2.55	2.58
DS	-	0.78	0.78	0.79

* Results obtained from the previous work [17]. Calculations were made per one PPU using the resonances from methoxyl groups as an internal standard [17].

## Data Availability

Data sharing not applicable.

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
