# Peer review of "Oxyalkylation of Lignoboost™ Kraft Lignin with Propylene Carbonate: Design of Experiments towards Synthesis Optimization"

_materials, 2022, doi:10.3390/ma15051925_

Round 1

Reviewer 1 Report

Please find the enclosed file.

Reviewer 2 Report

In this manuscript, response surface methodology was used to optimize the conditions for the preparation of lignin-based polyol. The effect and interaction of process variables on responses was investigated to obtain optimal values of hydroxyl number (IOH) and viscosity of lignin-based polyol to be used in polyurethane formulations of rigid foams and adhesives. This paper is interesting. However, more characterizations should be performed to validate the results of RSM optimization. In my point of view, the manuscript can be improved and accepted after major revisions. The suggestions are listed below.

(1) The authors are suggested to pay more attention to the usage of English grammar and tense in this paper.

(2) The graphs in the paper should be revised to make them more readable.

(3) Some corresponding characterizations are suggested to be performed to validate the resultant products prepared under the optimized process conditions.

(4) The format of the references should comply with the requirements of the journal.

Reviewer 3 Report

This manuscript deals with the effect and interaction of process variables for the oxyalkylation of Lignoboost™ kraft lignin with propylene carbonate by the full factorial modelling and response surface methodology. The work is interesting and significant, and the results are detailed and good. But the manuscript still needs some improvement.

  1. The pixels of all the pictures in this article are very low, it is very unclear. Please raise the pixels.
  2. The oxyalkylation mechanism of kraft lignin with propylene carbonate should be elaborated at length, preferably with mechanism diagram.
  3. Lots of basic characterization is missing, for example, FTIR, XPS, etc. before and after oxyalkylation.

Round 2

Reviewer 1 Report

NA

Reviewer 2 Report

The revised manuscript is suggested to be further published in Materials